# Unveiling Transformers with LEGO: a synthetic reasoning task

## Abstract

We propose a synthetic reasoning task, LEGO (Learning Equality and Group Operations), that encapsulates the problem of following a chain of reasoning, and we study how the Transformer architectures learn this task. We pay special attention to data effects such as pretraining (on seemingly unrelated NLP tasks) and dataset composition (e.g., differing chain length at training and test time), as well as architectural variants such as weight-tied layers or adding convolutional components. We study how the trained models eventually succeed at the task, and in particular, we manage to understand some of the attention heads as well as how the information flows in the network. In particular, we have identified a novel *association* pattern that globally attends only to identical tokens. Based on these observations we propose a hypothesis that here pretraining helps for LEGO tasks due to certain structured attention patterns, and we experimentally verify this hypothesis. We also observe that in some data regime the trained transformer finds "shortcut" solutions to follow the chain of reasoning, which impedes the model's robustness, and moreover we propose ways to prevent it. Motivated by our findings on structured attention patterns, we propose the LEGO attention module, a drop-in replacement for vanilla attention heads. This architectural change significantly reduces Flops and maintains or even *improves* the model's performance at large-scale pretraining.

## 1 Introduction

The deep learning revolution is about training large neural networks on vast amount of data. The first field transformed by this methodology was computer vision, crucially leveraging the convolutional neural network architecture LeCun et al. (1989); Krizhevsky et al. (2012). More recently natural language processing was revolutionized by the Transformer architecture Vaswani et al. (2017). Transformers are designed to process input represented as "set of elements" (e.g., the words in a sentence with their positional encoding). This is of course an incredibly generic assumption, and thus Transformers can be applied to a wide variety of tasks, including vision Dosovitskiy et al. (2021), reinforcement learning Chen et al. (2021a), and protein structure prediction Rives et al. (2021); Jumper et al. (2021) among others, or even jointly across domains to produce generalized agents Reed et al. (2022). In fact, learning with Transformers is rapidly becoming the norm in deep learning.

Transformer models display excellent performance on the standard criterion "training error/test error" (e.g., for masked language prediction or translation). However, what makes them particularly noteworthy, is that large-scale Transformer models seem to exhibit unexpected emergent behaviors, such as basic reasoning ability Thoppilan et al. (2022); Brown et al. (2020); Chowdhery et al. (2022); Du et al. (2021); Rae et al. (2021); Hoffmann et al. (2022); Smith et al. (2022); Zhang et al. (2022); Wei et al. (2022); Nye et al. (2022), excellent fine-tuning performance Hu et al. (2022); Thoppilan et al. (2022); Nye et al. (2022); Rae et al. (2021); Polu et al. (2022), or zero-shot learning Brown et al. (2020); Chowdhery et al. (2022); Du et al. (2021); Rae et al. (2021); Hoffmann et al. (2022); Smith et al. (2022); Zhang et al. (2022). Currently, there is a remarkable community effort towards *at-scale* experimental investigation of Transformers, essentially trying to find out what such models can do when they become large enough and are trained on large/diverse enough datasets. The successes are striking and capture the imagination Brown et al. (2020); Ramesh et al.

(2022). Yet, for all of these wonders, there is very little *understanding* of how these models learn, or in fact what *do* they learn. Answering such questions in the *at-scale* experiments is particularly challenging, as one has little control over the data when hundreds of billions of tokens are harvested from various sources. In this paper, we propose to take a step back, and try to understand how learning occurs and what is being learned in a more controlled setting that captures important aspects of "reasoning".

The benefit of such a controlled setting is that we can try to understand some of the most pressing questions in learning with Transformers, particularly around (i) the architecture and (ii) the importance of training data. For (i) we probe the role of multiple heads and depth, and we show that we can successfully understand them in our controlled setting. For (ii) we investigate how much the dataset composition matters, as well as how pretraining on merely vaguely related tasks makes fine-tuning successful. In turn, these insights can guide our thinking for large-scale experiments, and we give some of the lessons learned below. In particular, our insights crystallize into an architectural change to BERT for faster inference with matching or even better performance (Section 5).

## 1.1 LEGO: A synthetic reasoning task

Core components of reasoning include the ability to *associate* concepts, and to *manipulate* them. We propose a simple task that captures these two aspects, which we call LEGO (Learning Equality and Group Operations). In LEGO, the input describes a sequence of *variable assignments* as well as *operations* on these variables by a fixed (mathematical) group. One needs to be able to deal with both long-range assignments (the same variable appearing in different parts of the input should be viewed as a being *equal* to same quantity), as well as short-range operations (describing what group element is applied to which variable). A key parameter of an input sequence will be its length, which is proportional to the number of sequential reasoning steps one has to do in order to resolve the value of each variable. We will mostly train with a fixed sequence length (say 12). We often provide supervision only on part of the sequence (say the first 6 variables). We do so in order to test the generalization capabilities from smaller length sequences to longer length sequences without introducing potential errors due to the positional encoding in Transformers.

## 1.2 Some takeaways

In LEGO, we are interested in both *classical generalization* (i.e., training and test distribution are the same) and *out-of-distribution generalization*. For the latter we focus on distribution shifts that vary the length of the chain of reasoning, and thus we refer to this type of generalization as *length extrapolation*. Specifically, the setting for length extrapolation is to train with supervision on shorter sequence lengths (e.g., supervision on only the first 6 variables) and test on a long sequences (e.g., accuracy computed on 12 variables). A summary of our empirical observations is as follows:

1. First, classical generalization happens reliably for all architectures and data regimes.

2. More interestingly, length extrapolation seems to depend on architectural/data composition choices. Specifically, BERT-like models without special data preparation do *not* extrapolate to longer sequences, while other models like ALBERT, or BERT with carefully selected data (such as diverse sequence lengths, or pre-trained BERT) *do* extrapolate.

3. The extrapolating models all seem to evolve attention heads dedicated to either *association* (long-range identity matching) or *manipulation* (short-range operations). We provide evidence that pre-trained BERT (which is pre-trained on a seemingly unrelated dataset) generalizes because it has learned such heads.

4. The non-extrapolating models seem to solve the classical generalization problem using a certain shortcut-like solution, whereby using the specificity of the group operations they are able to jump to the end of the chain of reasoning, and then complete the rest of the variables by following the reasoning both from the start *and* the end of the chain.

We interpret our findings as follows:

(i) Classical generalization can be a deceptive metric, as there might be unexpected ways to solve the problem. This is famously related to the issue of embedding machine learning systems with *common sense reasoning*. Namely, we hope that when an ML system solves a task, it does so in "the way humans do it", but of course, nothing guarantees that this will happen. Our findings are consistent with the current methodology of increasing the diversity of the training data, which seems crucial for generalization.

(ii) ALBERT-like models, where a layer is repeated several times, seem to be an ideal structure for problems that could be described algorithmically as a "for loop" (as is the case with following a chain of reasoning). Indeed we find that ALBERT extrapolates in data regimes where BERT does not, clearly separating these two architectures.

(iii) The success of pretraining/fine-tuning in vastly different tasks might actually come from a "simple" better initialization, rather than complex knowledge encoded during pre-training.

(iv) The interplay between short-range (close-by information in a sentence) and long-range (the same concept appearing in different places in the sentence) is relevant more broadly than in our synthetic task. We observe that the networks effectively learn to deal with short-range/long-range information by implementing specific attention patterns. This motivates us to study a new LEGO attention architecture, and we show it matches or even outperforms its baseline on the large-scale pretraining but with significantly less computational cost.

## 1.3  Related works

In Zhang et al. (2021), the PVR (Pointer Value Retrieval) task is introduced, with a similar high-level goal to ours in introducing the LEGO task, namely to study how neural networks learn to reason in a controlled setting. In a PVR task, part of the input indicates another part of the input where a function of potentially varying complexity has to be computed. Like us, they use distribution shift to investigate how various network architectures learn this task, and they observe that networks can learn the task at hand ("classical generalization") yet fail to extrapolate to mild distribution shift. They then ask the following questions: "Are there architectural changes that can enforce better priors and withstand distribution shift? Can novel learning objectives prevent these adversarial correlations? Progress on these questions holds promise for greater robustness."

Our study attacks these questions directly in the context of the LEGO task (e.g., ALBERT versus BERT, and training set composition investigations), and our preliminary results indicate that this is indeed a fruitful direction to obtain better models in some aspects (e.g., more interpretable). Other examples of recent synthetic benchmark with a similar philosophy include SCAN (Simplified version of the CommAI Navigation) Lake & Baroni (2018), CFQ (Compositional Freebase Questions) Keysers et al. (2020), LIME Wu et al. (2021), PCFG SET Hupkes et al. (2020), and BONGARD-LOGO Nie et al. (2020). In SCAN for example, one has to "translate" a command of the form "turn left twice and jump" into a sequence of actions "LTURN LTURN JUMP" (see Patel et al. (2022) for more recent progress on this dataset). Again, similarly to the PVR tasks, these works focus on understanding generalization (in these cases, *compositional generalization*). Another related line of works is on studying Transformers to recognize various formal languages, see e.g., Bhattamishra et al. (2020); Yao et al. (2021). A contemporary work Csordás et al. (2021) proposed modifications to Transformer architectures to achieve significantly better length extrapolation (other works studying this important class of distribution shifts include Anil et al. (2022)). As far as we know, none of these works try to probe the inner workings of the networks in the same depth as we do here. On the other hand, networks trained on real data are being extensively scrutinized, see for example Rogers et al. (2020) where they try to understand some of the attention heads of BERT (see also Saha et al. (2020b)). However, making sense of these real-data-trained networks is a daunting task, and a key contribution of ours is to show in a limited setting one can obtain a clearer picture of what Transformers learn.

The LEGO task is also naturally related to the growing literature on testing mathematical/coding abilities of Transformers (e.g., Saha et al. (2020a)), specifically the simpler tasks of checking the correctness of a proof (or simplifying one, such as in Agarwal et al. (2021) which studies simplification of polynomials), or executing code for a given input Chen et al. (2021b). It would be interesting to see if some of the insights

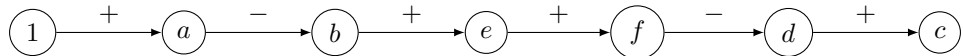

Figure 1: The graph representation of the sentence $a = +1$; $b = -a$; $e = +b$; $d = -f$; $c = +d$; $f = +e$

we derive in the present paper apply to currently challenging mathematical tasks such as MATH Hendrycks et al. (2021) and IsarStep Li et al. (2021).

There are an abundance of studies on attention heads that have identified the importance of local, convolutional, attention patterns Voita et al. (2019); Correia et al. (2019); Clark et al. (2019); Raganato et al. (2020); You et al. (2020). However, to the best of our knowledge, we are the first to demonstrate the importance of the association pattern that globally attends to identical tokens, thanks to the LEGO task.

## 2  Learning equality and group operations (LEGO)

We propose the following synthetic task, which we call LEGO. Let $G$ be a finite (semi)group acting on a finite set $X$, and denote $g(x)$ for the action of $g \in G$ on $x \in X$. We define a formal language using the symbols from $G$ and $X$ as well as symbols from a finite alphabet $A$ which we refer to as the *variables*. A sentence in our formal language is made of clauses separated by a semi-colon. A clause is of the form $a = gx$ with $a \in A$, $g \in G$ and either $x \in X$ or $x \in A$. If $x \in X$, such a clause means that the variable $a$ is assigned the element $g(x) \in X$. On the other hand if $x \in A$ and the variable $x$ was assigned an element $y \in X$ through another clause (or chain of clauses) in the sentence, then the clause $a = gx$ assigns variable $a$ to the element $g(y) \in X$. The task's goal is to take in input a sentence with a fixed number $n$ of clauses, given in an arbitrary order, and to output the assigned element to each variable that appear in the sentence (the formal language will have a further restriction that ensures that each variable is assigned one and only one element).

We can view a sentence as a directed graph on the vertex set $X \cup A$ with labelled edges as follows: a clause $a = gx$ corresponds to a directed edge from the vertex $x$ to the vertex $a$, and the edge is labelled with $g$. We restrict our attention to sentences corresponding to a line graph directed away from some fixed root vertex $r \in X$, and whose non-root vertices are all in $A$, see Figure 1 for an example. In particular such sentences are "consistent", meaning that a variable is assigned a unique element (the assignment is obtained by simply "following the chain").

**Task 1.** The most basic instantiation of LEGO is when $G$ is the unique group of 2 elements acting on a set $X$ also of 2 elements, that is $G = \{+, -\}$ and $X = \{1, -1\}$. Our sentences thus consists of $n$ clauses of the form $a_i = \pm a_{i-1}$, where $a_i \in A$ for $i = 1, 2, \ldots, n$ and $a_0 = 1$ (we fix $r = 1$). Note that in this case our formal language has well over a billion unique valid sentences when $n \geq 10$. Example of a sentence with $n = 6$ is (see Figure 1 for the graph depiction): $a = +1$; $b = -a$; $e = +b$; $d = -f$; $c = +d$; $f = +e$. Our task's goal is to report the elements or values from $X$ assigned to the variables appearing in the sentence. In the above example, assignments for variables $a, b, c, d, e, f$ are $1, -1, -1, -1, 1, 1$.

**Task 2.** One can think of Task 1 as the case of LEGO for the permutation group on $N = 2$ elements (acting on itself). Our second task will correspond to $N = 3$, which is qualitatively different since the permutation group on 3 elements is non-abelian.

We will focus on Task 1 in the main paper and include in Appendix E experiments on this Task 2. Our training and test data for the task consists of $n$ length chains as described above with the order of clauses in the sentence randomized. A sample input sentence to a transformer looks like `[BOS] j=-f; f=-b; y=+t; o=+e; d=+y; v=+d; h=-o; b=-i; i=+1; t=+l; e=-j; l=-h; [EOS]`. See appendix for further data generation details.

## 3  Transformers for LEGO

We apply transformer models in the token classification pipeline to predict the assignments of the variables in the input sentence, depicted in Figure 2. To evaluate the out-of-distribution generalization (referred to simply as generalization), we introduce the notation of $n_{tr} \leq n$, such that during training, supervision is provided only on the first $n_{tr}$ clauses (first in the graph representation of the input sentence). We mainly focus on BERT Devlin et al. (2018) and ALBERT Lan et al. (2019) architectures. These two models are

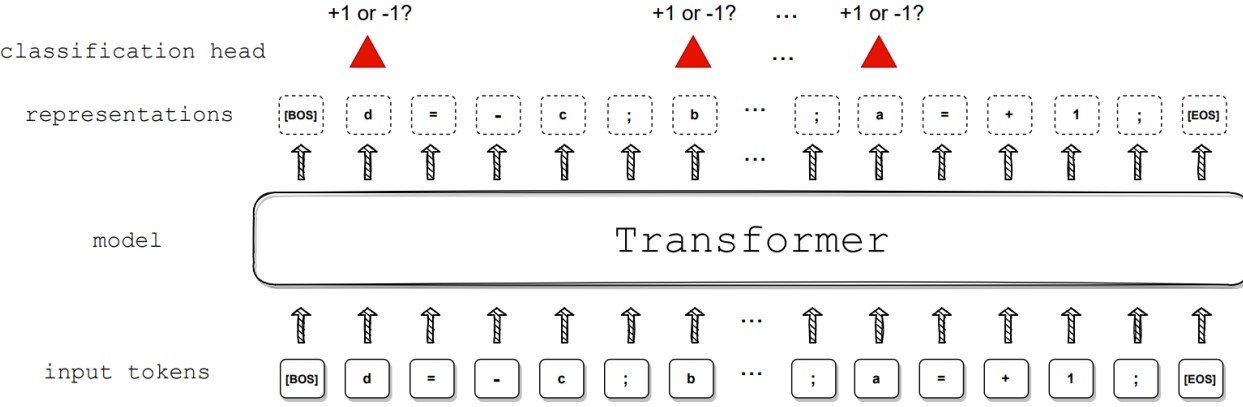

Figure 2: Illustration of a transformer model applied to LEGO Task 1 on input sentence d=-c; b=-a; c=+b; a=+1;. We apply a linear classification head to the output representations of each clause's first token to generate predictions for the variables assignment.

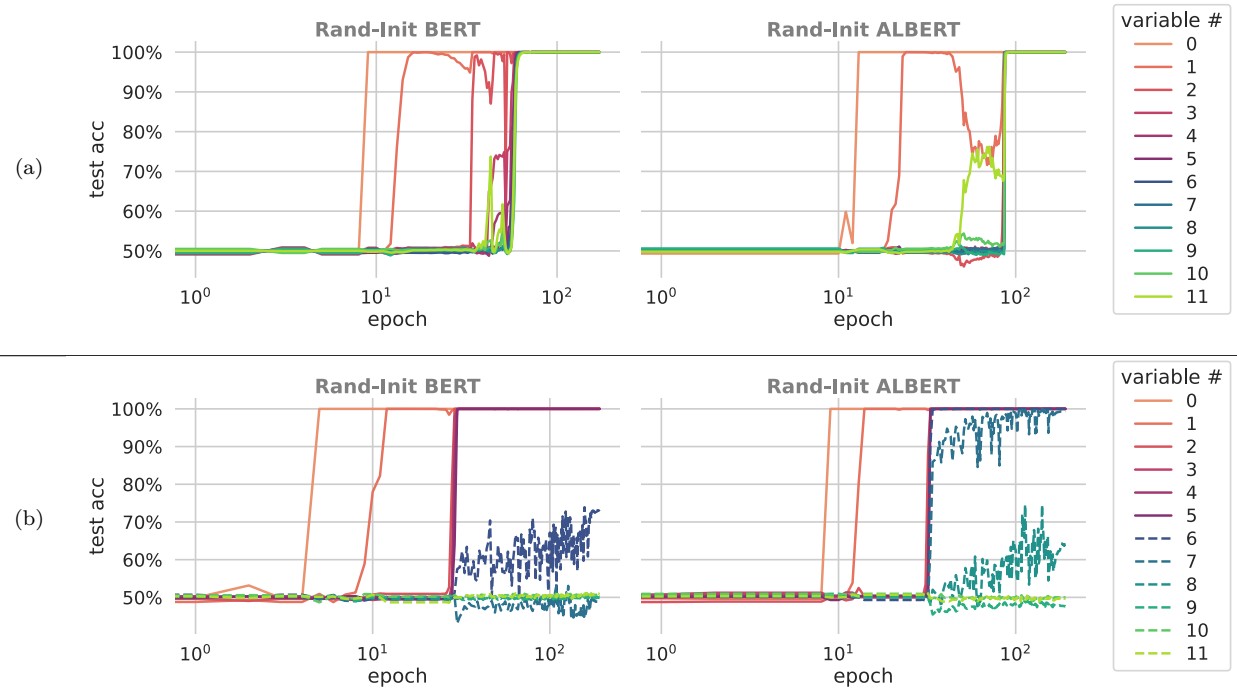

Figure 3: Solving LEGO (Task 1) using BERT and ALBERT, trained from random initialization. Each curve corresponds to the test accuracy of a single variable appearing in the sentence over the course of training. The variable numbers in the legend are their position in the reasoning chain (or graph representation) of the input sentence, rather than the position in the sentence itself. For example, on the input sentence: b=-a; d=-c; c=+b; a=+1;, variable #0 is a, #1 is b, #2 is c, and #3 is d. Top a): models are trained to fit all variables ($n = 12, n_{tr} = 12$). Bottom b): models are trained to fit the first 6 variables but test on all 12 variables ($n = 12, n_{tr} = 6$). Dashed curves represent variables not supervised during training.

representative large transformer architectures for NLP tasks, and we observe they exhibit intriguing behavior difference on our tasks which we will detail in Section 4. See appendix for training hyper-parameters and dataset construction details.

In Figure 3, we report initial results on LEGO with $n = 12$ and $n_{tr} = 6, 12$. Both BERT and ALBERT are able to achieve good classical generalization, while only ALBERT appears to generalize even to slightly

longer sequence length. We observe similar behavior across different lengths of inputs too. This suggests that classical generalization might be a deceptive metric to evaluate learning of true logic/reasoning tasks. Motivated by these initial results, in the next section we focus on breaking down the learning dynamics of BERT and ALBERT for the LEGO task towards carefully understanding their strengths and weaknesses.

# 4 Unveiling Transformers with LEGO

## 4.1 BERT vs. ALBERT: Iterative reasoning in iterative architectures

A salient feature of many reasoning tasks is an iterative component, meaning they can (or must) be solved by sequentially repeating certain operations. In this section, we use LEGO to study and compare Transformer architectures through the lens of iterative reasoning.

A natural solution to LEGO—and arguably the go-to solution for a human—is to implement a "for-loop", where each iteration resolves one step in the reasoning chain. The iteration could look for the next unresolved variable token whose value could be resolved in one step. Iterative Transformer architectures such as ALBERT and Universal Transformers Dehghani et al. (2018), where the weights are shared across different layers, inherently implement a for-loop with a number of iterations equal to the number of layers. If the model manages to learn to implement one such iteration during training, the network would immediately be capable of performing length extrapolation. If this indeed occurs, it would point to a clear advantage of ALBERT over BERT in our setting. This leads to the following questions.

### Q1. Do iterative architectures indeed exhibit better length extrapolation?

The bottom plots of Figure 3 display the length extrapolation result for BERT and for ALBERT. They show the clear advantage of recurrence: While the non-iterative BERT achieves only somewhat better-than-random accuracy for one variable (#6) beyond the ones accounted for during training (#0- -#5), the iterative ALBERT reaches near-perfect accuracy on two additional variables (#6 and #7), and nontrivial accuracy on the third (#8). These results clearly support that iterative architectures do generalize better in the iterative LEGO reasoning task.

### Q2. Does the ALBERT architecture actually implement the for-loop?

To a lesser extent, Figure 3 also hints at a positive answer to Q2. Observe that ALBERT exhibits length extrapolation to variable #6 immediately (in terms of epochs) as soon as it fits the training variables (#0 – #5), whereas for BERT, the corresponding plot (#6) climbs gradually even after the training variables are predicted perfectly. This suggests that once it manages to learn the operations required for one step of reasoning, it can immediately implement those operations over a few more iterations not required in training.

In order to gain stronger evidence, we measure the dependence between the location of a variable token in the chain and the layer in which its value is typically resolved. To this end, given a trained model, we train one linear classifier per layer which predicts the value of a variable token based only on its token representation at the corresponding layer (without using other information), while keeping the original model unchanged. This allows us to gauge the rate of information percolation along the reasoning chain in terms of layers per reasoning step. If the model indeed implements a for-loop in its forward pass, one expects a linear relationship between the number of layers and the number of reasoning steps already completed. We visualize in Figure 4 the test accuracy of prediction as a function of the layer in the network and depth in the chain. While not perfectly linear, the relation clearly looks closer to linear in ALBERT, suggesting that the ALBERT model has an inductive bias towards learning to implement the "natural" for-loop with its forward pass.

### Q3. How can we incentivize models to learn iterative solutions?

We attempt to incentivize the model to implement the "natural" for-loop solution. We rely on the observation that if each iteration of the for-loop simply percolates the information one more step (assigning a value to the next variable), then adding more layers with the same weights should not affect the output, and in fact, one should be able to read out the output of the calculation from any layer of the neural network, as long

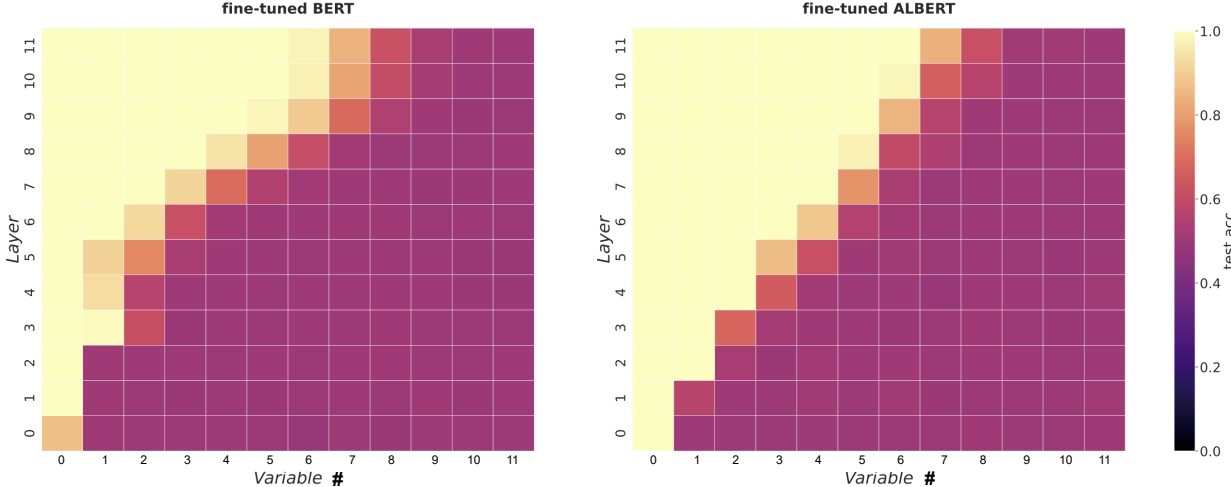

Figure 4: Visualization of information percolation within the fine-tuned models. The color indicates the test accuracy of the probing classifier at each layer. Brighter is higher. We observe ALBERT's information percolation is linear than BERT's, which implies ALBERT is biased towards learning a for-loop.

as its depth exceeds the length of the chain. With this observation in mind, we train a ALBERT model with stochastic depth Huang et al. (2016). We uniformly sample depth between 6 and 12 per batch during training while fixing it at 12 during test. Figure 5 shows a clear improvement in generalization to longer lengths using stochastic depth.

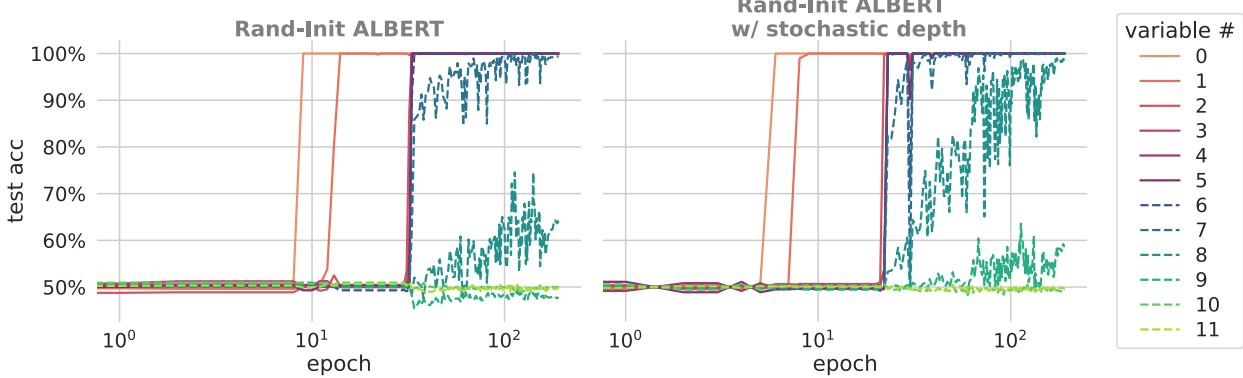

Figure 5: Generalization of ALBERT trained with stochastic depth. The stochastic depth improves the length extrapolation to longer sequence lengths.

## 4.2 Rand-Init vs. Pretrained: Structural advantages from pretraining

Pretraining large models has emerged as a prominent and highly successful paradigm in large-scale deep learning. It advocates first training the model on a large dataset to perform a generic task, followed by task-specific fine-tuning on the task at hand. Our goal here is to use LEGO as a testing ground for this paradigm. To this end, we compare (a) training the BERT architecture for LEGO from random initializations to (b) fine-tuning the standard pre-trained BERT model to solve LEGO. Figure 6 (left and center plots) shows that pretraining helps generalization in LEGO dramatically: the pre-trained model generalizes to unseen sequence lengths (the dashed plots) much better, and within a far smaller number of epochs, than the randomly initialized model.

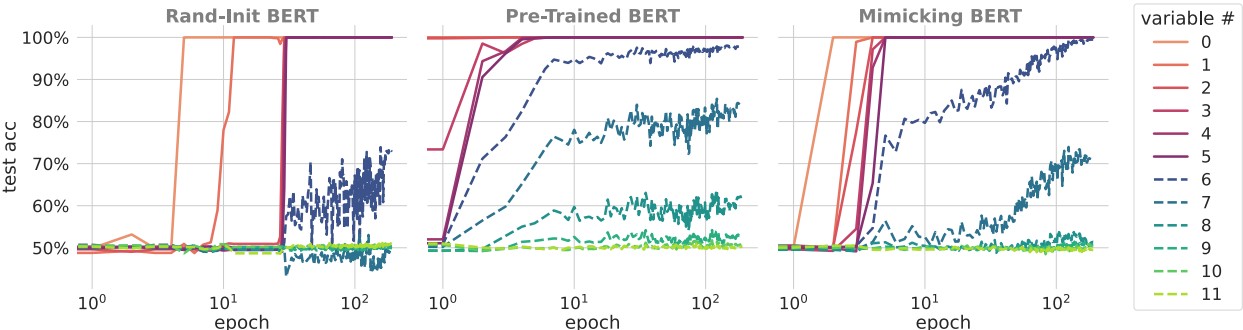

Figure 6: Pretrained BERT exhibits significant performance advantages over its Rand-Init counterpart, while the mimicking procedure (a simple initialization scheme we describe below) heads closes the gap.

### 4.2.1  Why does pretraining help in LEGO?

One simple explanation is that pre-trained BERT is already aware of the semantics of tokens like '=' or '-'. We have easily ruled out this possibility, by replacing those tokens with arbitrary ones that do not encompass the same semantics; this does not affect the performance of pre-trained BERT. A more intriguing explanation pertains to the attention mechanism itself. At its basis, LEGO requires two fundamental types of information transfer:

- **Association:** encoding long-range dependencies that transfer a value between two occurrences of the same variable. For example, if the input contains the two clauses "$a = +1$" and "$b = -a$" (with arbitrary separation between them), the architecture must associate the two occurrences of the variable $a$ in order to correctly set $b$ to $-1$.
- **Manipulation:** encoding short-range dependencies of transferring a value from the right-hand to the left-hand side of the clause. For example, to successfully process the clause "$b = -a$", the architecture must associate these particular occurrences of $a$ and $b$ with each other, in order to transfer the value of $a$ (after applying to it the group element $-1$) into $b$.

Association corresponds to a purely global attention pattern, completely reliant on the *identity* or *content* of the tokens and oblivious to their *positions* in the input sequence. Manipulation, in contrast, corresponds to a purely local attention pattern, where nearby positions attend to each other.

It is natural to ask whether they are indeed manifested in the pre-trained model's attention heads in practice. Indeed, Fig. 7 shows two exemplar attention heads of pre-trained BERT on an input LEGO sequence without any fine-tuning. The right head clearly depicts association: each token attends to all other occurrences of the same token in the input sequence. This motivates us to propose the following hypothesis:

**The advantage of pre-trained models on LEGO can be largely attributed to the association and manipulation heads learned during pretraining.**

Note that merely the existence of the heads does not fully validate the hypothesis yet. To rule out other factors, we carefully design controlled experiments to test this hypothesis in the section below.

### 4.2.2  Verifying the hypothesis with Mimicking

To test this hypothesis, we conduct the following *mimicking* experiments.

**Mimicking BERT**  We 'initialize' certain attention heads to perform association and manipulation, without access to pretraining data. We achieve this by specifying the target attention matrices (one for association and one for manipulation), and training the model on random data to minimize a "mimicking loss" that measures how well the actual attention matrices at every layer match the target matrices. The precise mimicking loss and training protocol are specified in the Appendix B.3. The rightmost plot in Figure 6 shows that BERT with mimicking initialization attains significant advantage in generalization over randomly initialized BERT,

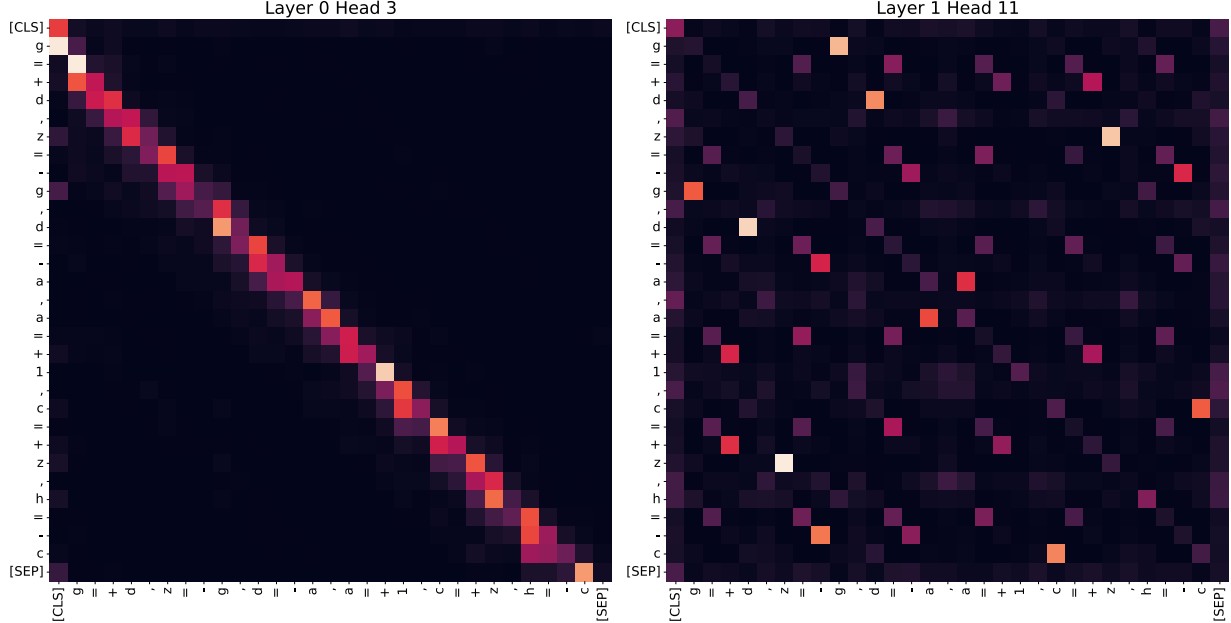

Figure 7: Visualization of two representative attention maps from a pre-trained BERT model not yet fine-tuned on LEGO. A complete visualization of all attention patterns of the pre-trained BERT is in Appendix F. On the LEGO input sequence, certain heads implement local, convolution-like manipulation operators (left), while some others implement global, long-range association operators (right), i.e. token 'z' attends to the other appearance of 'z', token '=' attends to all the other '='s. Note that the sample input sequence is presented in the reasoning chain order for visualization purposes only.

despite not being pre-trained on any real data (and thus not having learned to "reason"). This confirms that much of the advantage of pre-trained BERT stems from having learned these information transfer patterns.

### 4.3 Shortcut solutions and their effect on generalization

As discussed in Section 4.1, a natural solution to LEGO is to resolve variables iteratively by the order of their depth in the chain. Surprisingly, we find that the Rand-Init BERT and ALBERT models first learn a "shortcut" solution: they immediately resolve the *last* variable in the reasoning chain, perhaps by counting the total number of minus signs. Indeed, the last variable can be easily identified as it appears only once whereas every other variable appears twice, and its value is fully determined by the parity of the number of minus signs. This behavior is observed in Figure 3a: the randomly initialized models are trained to fit all 12 variables: the last variable (#11, indicated by the brightest green curves) improves earlier than *almost* all other ones.

This behavior may be related to the well-observed phenomenon of *spurious features*: a model succeeds in training not relying on any actual features of cows and circumventing the intended solution McCoy et al. (2019); Srivastava et al. (2020); Gururangan et al. (2018); Nguyen et al. (2021).

We use LEGO as a case study of shortcut solutions and their effect on generalization. Instead of training the model to fit the first six variables (as in bottom Figure 3 in Appendix), we train it to fit the first five (#0–#4) and the last variable (#11). This allows us to measure length extrapolation (to #5–#10) in a setting where models can learn the shortcut. The results show significantly degraded performance, implying that shortcut solutions impede generalization. We then study ways to prevent models from learning them, by pretraining and mimicking. The full section appears in Appendix A.

## 5    LEGO Attention: faster and better

Our analysis in Section 4.2 reveals that the advantage of the pre-trained BERT model on LEGO originates from two specific types of attention structures emerging from pre-training — the association and manipulation patterns. A quick examination of all the attention heads depicted in Appendix F suggests that there is one more clearly identifiable attention pattern: broadcasting on the `[CLS]` token or the `[SEP]` token (sometimes both). Namely, it 'broadcasts' the value inside the special tokens to the others. Even though `[CLS]`, `[SEP]` play no role on LEGO per se, they are vital to the pretraining objective as well as many downstream tasks. Thus the broadcasting attention pattern is presumably important for many real-life NLP tasks beyond LEGO. Association, manipulation, and broadcasting consist of a considerable portion of the pre-trained BERT's attention heads, and they are so structured that we simulate them efficiently.

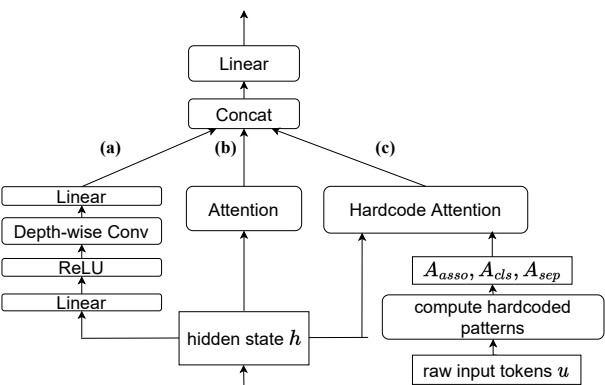

Figure 8: Our proposed LEGO attention consists of 3 pathways. BERT has pathway (b) only; the LEGO v0 attention module has (a) and (c); the LEGO v1 attention has (a), (b), and (c). See Appendix C.

**LEGO Attention:**    For the association, manipulation, and broadcasting heads, we can efficiently construct the sparse attention matrix based on the input token IDs only, without learning $Q$ and $K$ or the expensive attention probability computation. For manipulation maps, due to their intrinsic locality, we decide to implement them directly with temporal convolutions (along the time dimension). For the other global maps, given a *raw* input sequence of $T$ tokens, $u_1, u_2, \ldots, u_T \in \mathbb{N}$, we manually construct the association and broadcasting maps $A_{asso}, A_{cls}, A_{sep} \in \mathbb{R}^{T \times T}$ such that $(A_{asso})_{ij} = \mathbf{1}\left[u_i = u_j\right]$, $(A_{cls})_{ij} = \mathbf{1}\left[u_j = \texttt{[CLS]}\right]$, $(A_{sep})_{ij} = \mathbf{1}\left[u_j = \texttt{[SEP]}\right]$ where $\mathbf{1}\left[\cdot\right]$ is the indicator function which outputs 1 if the argument is true and 0 otherwise. In the end, we normalize them to have row-wise unit $\ell_1$ norm. Notably, the latter three patterns require no training (except for a value map for each layer) and are shared across all layers.

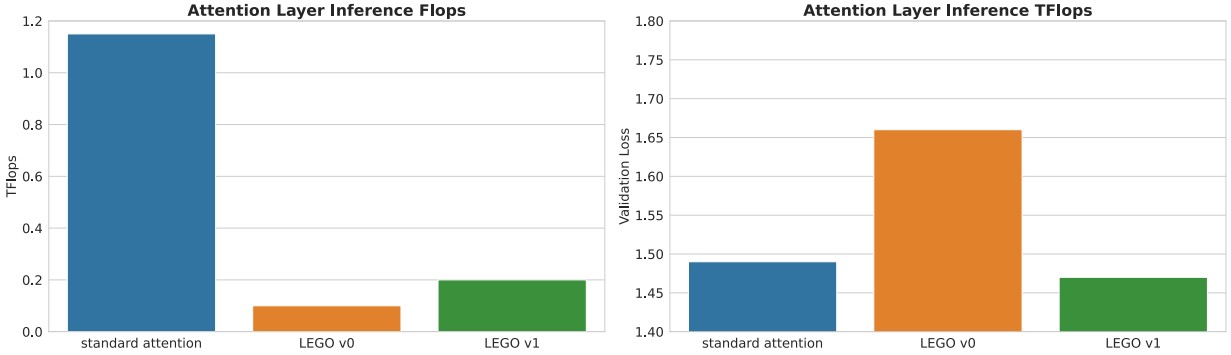

Figure 9: Comparison of inference Flops and model size. Flops are measured on a batch of 64 sequences of 512 tokens.

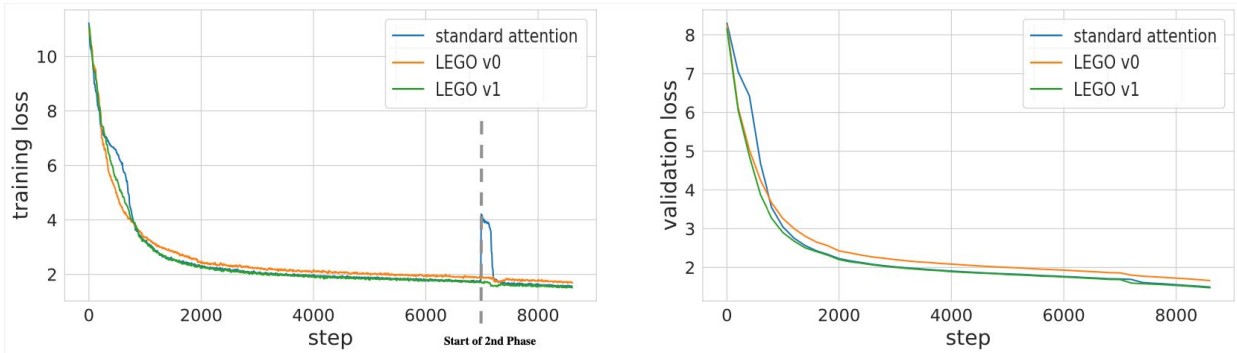

Figure 10: Training and validation performance on BERT pertaining task (Masked Language Modelling+Next Sentence Prediction). As a standard, the training sequence length increases from 128 to 512 around the 7k-th step, where the BERT training loss exhibits a sudden bump in response, while the LEGO v0/v1 models both exhibit resilience. The LEGO v1 model learns faster and (slightly) outperforms BERT in validation.

On the standard BERT pertaining benchmark, we compare the following three models: BERT-base model, LEGO v0 and v1 models. We use convolutional kernel size 21 for the latter two. In Figure 10, we show that the LEGO v0 model learns fast in the beginning but falls short later on. However, the LEGO v1 model not only reduces model size and accelerates inference, but also renders models that are extremely competitive with the base model in terms of the final performance of large-scale pertaining. We follow precisely the training pipeline and hyperparameters of Devlin et al. (2018). See Appendix C for architecture details of the LEGO v0/v1 models.

We observe that the LEGO v0 model learns faster but gradually falls short, while the LEGO v1 model achieves the best of both worlds: it learns faster at the beginning and achieves even (slightly) lower validation loss at the end. The LEGO v1 model's validation loss curve appears to be a lower envelope of the other two. The BERT/LEGO v0/v1 models achieve 1.49/1.66/1.47 final pertaining validation loss and 88.2/82.5/88.1 Dev F1 score on SQuAD v1.1 Rajpurkar et al. (2016). We leave comprehensive evaluations for future work.

**Difference from previous work: LEGO Attention is sparse and input-dependent.** Many previous studies have proposed to replace standard attention heads with structured, sparse patterns for speed Voita et al. (2019); Correia et al. (2019); Clark et al. (2019); Raganato et al. (2020); You et al. (2020), usually at the cost of inferier performance on standard tasks. The common drawback of previous sparse attention patterns is that they are input-independent, i.e. the patterns are fixed before seeing any input sequences. To the best of our knowledge, the association head of LEGO Attention is the first sparse *and* input-dependent attention pattern discovered from pretrained models themselves, which enables it to provide speedup while boosting performance.

## 6 Conclusion

In this work, we study Transformers by constructing LEGO, a controllable synthetic logical reasoning task. With LEGO, we have gained insights into their inductive bias, the role of pertaining, etc. Based on these insights, we proposed the LEGO attention mechanism which both accelerates inference and leads to comparable or even better performance. There are many important attention heads beyond just manipulation and association, and their roles remain to be discovered. We believe LEGO will continue to deepen our understanding on Transformers' inner working and to inspire better algorithms/architectures for tasks beyond LEGO.

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
