# OpenReview forum: "Unveiling Transformers with LEGO: a synthetic reasoning task"
_TMLR — Rejected by TMLR_

### Review · Reviewer_mQE9 · 2023-05-02

**Summary Of Contributions:**

This paper contributes,
- LEGO -- a novel synthetic dataset for studying LLMs. The main LEGO task that is studied is to predict variable assignments in a chain of dependencies where each assignment is of the form $x={+,-}y$. This allows controllable study of LLMs on both long-term (current state of variables based on the past) and short-term (variables and operators in scope) dependencies.
- Empirical analysis of the attention mechanism in BERT-style LLMs. The authors show that ALBERT is a better fit for iterative reasoning due to replicating layers.
- Empirical comparison between different initialization approaches, i.e., rand-init, language pretraining, and mimicking pretraining. The authors show that by simply pretraining a LLM to mimic a predefined attention pattern, the performance is on-par with language pretraining. While predefined attention pattern is manually devised in this setting, it proposes an interesting perspective that a good prior on attention weights is enough to achieve competitive results on LEGO.
- A hybrid attention mechanism -- convolution, self-attention, manual attention -- based on the specific long-term and short-term structures. Convolution and self-attention captures short-term and long-term structure while manual attention gives a bias based on token matching. This attention mechanism achieves a similar perplexity on holdout validation set compared to standard attention.

**Audience:**

Yes

**Broader Impact Concerns:**

There are no concerns.

**Claims And Evidence:**

Yes

**Requested Changes:**

Please see above.

**Strengths And Weaknesses:**

**Strengths**

- Controllable tasks such as LEGO that highlight interesting patterns/findings in complex architectures are very useful. LEGO specifically captures two important characteristics -- long-term and short-term dependencies -- in a bounded space.
- Understanding architectural choices for reasoning is important. Comparison between ALBERT and BERT shows layer replication is important for better reasoning.
- Comparison across different initialization schemes, especially mimicing objective, shows while language pretraining can encompass many  sub-tasks, simpler objectives can work really well. This is useful in understanding pretraining datasets and objectives for different downstream tasks.

**Weaknesses**

Mainly, there are several claims in the paper that needs clarification.

- You claim that in Figure-4, ALBERT looks much linear compared to BERT. To me, it seems more like the slope is almost the same but the bias term for BERT is higher than ALBERT. I think fitting a simple affine function to the results would clarify.
- While the LEGO task could be solved using a for loop, is that the only way to solve it? My main objection is to the claim that ALBERT has an inductive bias towards implementing a for-loop due to this reasoning.
- In Appendix-D, you explain that training on longer chains give better OOD generalization, mainly because of having sufficient information in the data. This makes me wonder about the "long-term dependency distribution" in the LEGO task. If a chain is reset very frequently, meaning that two parts of a chain are independent, this could easily be just two independent sequences. In that case, using longer chains could just mean that you have more data; this could also explain why you get better results with increasing sequence lengths. Could you please clarify with a plot?
- Given that shortcut solution is the main problem in LEGO and it is also easily alleviated via pretrained models, I wonder if Task-1 is overall a challenging benchmark. Could you please discuss?
- The impact of tokenization is not mentioned in the paper but I think that is crucial, especially for math problems. This also has a significant impact on the LEGO attention as hardcode attention ensures the same variables, operators, etc. are matched.
- The sentence and graph in Figure-1 don't match.

---

> ### Author Response · Authors · 2023-06-10
> **Response**
>
> @ “In Figure-4, the slope is almost the same”:
>
> The R-Squared values for linear fits on the left and right subplots are 0.96 vs 0.99, suggesting that the right hand side plot (ALBERT) is more linear.
>
> @ “While the LEGO task could be solved using a for loop, is that the only way to solve it?”
>
> It is not and this is precisely why we investigate further in “Q2. Does the ALBERT architecture actually implement the for-loop?” in section 4.1. Our empirical evidence in Figure-4 shows that ALBERT’s solution is closer to a for-loop than BERT’s.
>
>
>
> @ “Given that shortcut solution is the main problem in LEGO, and it is also easily alleviated via pretrained models, I wonder if Task-1 is overall a challenging benchmark. Could you please discuss?”
>
>
>
> Note that the pretrained model (BERT/ALBERT) has been trained on billions of tokens for at least hundreds of GPU hours. Thus, the fact that they alleviate the shortcut issue does not mean our task is not challenging enough.
>
> Quite the opposite, our seemingly innocent task makes transformers trained from random-init fail means it is quite challenging. It also provides a lens to probe the functional benefits of pretraining, which we carefully anlayze in section 4.2.
>
>
>
> @ “the impact of tokenization is not mentioned in the paper”:
>
> In this work, we focus on the basic setting where each character in the LEGO sequence is tokenized into a standalone token. We will add a note to the revision.
>
>
>
> @ ”The sentence and graph in Figure-1 don't match.”:
>
> They do. In fact, in figure 1, that the ordering in the sentence is different from in the graph is INTENTIONAL. Note that in a LEGO sentence, the clauses are randomly shuffled.

---

### Review · Reviewer_kN6k · 2023-05-08

**Summary Of Contributions:**

This work introduces a diagnostic benchmark to better understand the compositional generalization of Transformers. In particular, this work compares BERT and ALBERT models on a synthetic task consisting of predicting a numerical assignment for each variable in a randomly ordered sequence of numerical associations such as mapping `“c = -b ; a = 1 ; b = a ; …”` to values `“1, 1, -1”` respectively for `a, b, c`.
Models are trained on sequences of 12 variables but with supervision on only the first 6. Models are then evaluated on all 12 variables.
Multiple settings are considered: (1) trained from scratch, (2) pre-trained + finetuned, and (3) trained from a smart initialization.

The experimental results show the following conclusions:
- Iterative architectures (or weight-sharing models) such as ALBERT extrapolate better than non-weight-sharing models such as BERT.
- Iterative models such as ALBERT seem to implement a “for-loop” like procedure in their weights.
- Training with a stochastic depth significantly improves extrapolation performance.
- Pre-trained models train faster and generalize better than models trained from scratch.
- A smart initialization of the attention matrices can partially explain why pre-trained models perform better in fewer training steps.

Overall I think this is a good paper that provides a lot of insights but that would benefit from some organization and further discussion of the presented results.


**Audience:**

Yes

**Broader Impact Concerns:**

No Broader Impact Statement required.

**Claims And Evidence:**

Yes

**Requested Changes:**

**Critical requests**:
- Clarify one aspect of the methodology (see Weaknesses above).
- More discussion of the results:
  - Comment on the iterative -vs- weight-sharing aspects of ALBERT
  - Can the linear relation between layer and variable number in ALBERT come from the equal split of compute resources rather than a “for-loop” like procedure? Are they the same?
  - Why are there strong improvements in Figure 5?
  - Comment on the performance difference between pre-trained BERT and Mimicking BERT.
- Show the benefits of the proposed LEGO attention with extrapolation results. These results can replace Figures 9 & 10 for instance.

**Minor requests**:
- In Section 4.1 / Q2: ALBERT performs well on variable #6 as soon as it performs well on variables 2-5 (~50 epochs) but not as soon as it performs on variables #0 and #1 which happens around epoch 10. This should be corrected in the manuscript.
- In related works, also mention CLUTRR (Sinha et al., 2019) and RuleTaker/ProofWriter (Clark et al., 2020; Tafjord et al., 2020) as datasets with similar compositional generalization philosophy.

**Suggestions to make the paper stronger but not required for acceptance**:
- Evaluate an autoregressive decoder model such as GPT2.
- Evaluate a 6 layer and 18 layer ALBERT and plot information percolation as in Figure 4.


**Strengths And Weaknesses:**

**Strengths**

The paper has a lot of insightful results listed in the summary section above.

---

The methodology is somewhat clear and easy to follow (except for one question - see weaknesses)

---

The compositional generalization and length extrapolation research topic are of great interest to the community as multi-step reasoning is often a limitation of large language models.

---

**Weaknesses**

What the paper calls “classical generalization” (same training and testing distribution) is rarely used in the literature to evaluate logic/reasoning tasks as it is not testing for reasoning. Hence the *finding* that “classical generalization can be a deceptive metric”, while being true, is not as insightful as the rest of the paper. This should probably be rephrased into a motivating point for the experimental setup used in the rest of the paper.

---

One aspect of the methodology is not entirely clear.
The paper mentions that models are trained on fixed sequence length (say 12 variables) but only give supervision to the first 6 variables. It is not clear how this is done. Is it by (1) having sequences with 12 variables during training but only giving labels to the first 6 variables in the reasoning chain or (2) by having sequences with only 6 variables during training and 12 variables during testing? I think that setup (1) is used but it should be more clearly stated. In addition, setup (1) is easier than (2) and does not *entirely* test for extrapolation.
The paper must discuss this distinction and either:
- motivate the decision of using method (1), or
- report additional results with the second setup.

---

Results are very briefly discussed. As a result, multiple follow-up questions arise (listed below).
For instance, Section 4.1 / Q3 presents significant improvements but the reason why is not discussed. There is no analysis in this section.

---

The LEGO attention in Section 5 is not well introduced. It is not well integrated with the rest of the paper. In addition, this method is not evaluated like all the previous experiments hence it is not clear if it is relevant. Lower validation loss curves do not necessarily imply better results, especially in highly controlled synthetic reasoning tasks like LEGO. I would suggest to either move that section to the appendix or to better integrate it with the rest of the paper by running the same evaluations as in Figures 3, 5, and 6.

**Follow-up questions**:

4.1/Q1: is it the iterative aspect of ALBERT that improves extrapolation or the fact that the model has weight sharing? It would be nice to discuss this by comparing results with different weight-sharing approaches or different models.

---

From Figure 4 it is particularly interesting to see that ALBERT seems to almost systematically use two layers to perform perfectly on each additional variable. Is this inherent to the difficulty of the task or is this because the model was trained with 12 layers on 6 variables and is thus good at splitting its compute resources equally? One experiment to test this would be to train the same model with 6 and 18 hidden layers and plot the same results as in Figure 4 to see if the model still requires 2 layers per variable or if it switches to 1 for the 6-layer experiment and 3 for the 18 layers experiment.

---

4.1 / Q3: why is stochastic depth training so effective?

---

Figure 6 shows that a smarter initialization of the attention layers significantly improves BERT extrapolation capacity. However it is not as effective as the pretrained model. Can the authors comment on that difference and suggest what else is captured by the pre-trained weights?

---

**limitation**:

Only encoder transformers are evaluated in this work, it would be interesting to see an evaluation on an autoregressive decoder model such as GPT2.

---

> ### Author Response · Authors · 2023-06-10
> **Response**
>
> @ “It is not clear how only giving supervision to the first 6 variables is done”:
>
> Indeed, the reviewer’s comment (1) accurately describes this procedure, which is “having sequences with 12 variables during training but only giving labels to the first 6 variables in the reasoning chain”.
>
> Our motivation for adopting (1) instead of (2) (training on sequences of 6 variables while testing on sequences) is simply that models trained with setup (2) will naturally fail due to lack of training of the position encodings of the longer sequences. Note that this caveat applies to both absolute and relative position encodings.
>
>
>
> @ “is it the iterative aspect of ALBERT that improves extrapolation or the fact that the model has weight sharing?”
>
> Indeed, we have experimental results in Appendix D.1 Effect of number of parameters which show that it is the iterative aspect instead of fewer parameters of ALBERT that improves extrapolation.
>
>
>
>
>
> @ “Can the authors comment on that difference and suggest what else is captured by the pre-trained weights?”
>
> Digging further into what the pretrained weights capture is an interesting and challenging task. Note that the X-axis in our plots is log-scale, thus the gap difference between mimicking init and pretrained weights may not be as large as it seems. However, we do believe there could be simple reasons such as norm of parameters, layer balancedness that our mimicking init does not capture yet.

---

### Review · Reviewer_ojhP · 2023-05-15

**Summary Of Contributions:**

The paper introduces LEGO, a synthetic reasoning task requiring a chain of reasoning, and investigates how Transformer architectures learn it. They examine factors like pretraining, dataset composition, and architectural variations. They identify a novel attention pattern and propose that pretraining benefits LEGO tasks due to structured attention patterns. They also address the issue of shortcut solutions and present the LEGO attention module as a more efficient replacement for vanilla attention heads, improving performance during large-scale pretraining.


**Audience:**

Yes

**Claims And Evidence:**

No

**Requested Changes:**

Results on different combinations of (n, n_{tr}, L) are required to claim length extrapolation fully.

There might be many different shortcuts except the one explained in the paper. For example, assuming the linear chain of the input, simply counting can solve the problem, meaning that one-layer attention can solve this task.

Repeating a similar process on different neural architectures and comparing it with the results from the transformer might be helpful.

I wonder if the accuracy (in Figures 3, 5, and 6) always goes to 100% with enough training epochs. If not, why?

Figure 9: the title of the right subfigure is wrong.


**Strengths And Weaknesses:**

My main concern is LEGO task is synthetic and too easy to demonstrate the transformer’s characteristics. For instance, RNN and CNN can do a perfect job on this task. LEGO is even solvable by an elementary rule-based program, so we will not use a transformer to solve LEGO in practice. The insights obtained from results and analysis may not extrapolate to other more complex reasoning tasks.

This work examines important recipes of transformer training, including pretraining, parameter sharing, and attention architecture. Although the design of the entire experiment looks reasonable, the results are not surprising enough.

---

> ### Author Response · Authors · 2023-06-10
> **Response**
>
> @ “LEGO task is synthetic and too easy, RNN and CNN can do a perfect job on this task”:
>
> We would like to emphasize that LEGO is NOT solvable by RNNs or CNNs. Intuitively, RNNs fail at LEGO because of the random ordering of the variables in the LEGO task, and CNNs fail because of the lack of global/non-local operations in their architecture.  This is consistent with our experimental observation as well: RNNs and CNNs can barely predict the values of the 2nd variable along the chain order in the LEGO task with 12 variables, thus far from solving it.
>
>
> @ “Results on different combinations of (n, n_{tr}, L) are required to claim length extrapolation fully”:
>
> Note that we have experimental results precisely on this in Appendix D.
>
>
>
> @ “For example, assuming the linear chain of the input, simply counting can solve the problem, meaning that one-layer attention can solve this task.”
>
> Note that this solution indeed coincides with the short-cut solution we mentioned in the paper.

---

### Decision · Action_Editors · 2023-07-07

**Recommendation:** Reject

**Comment:**

This work provides interesting insights and would be of interest to the community. However, even after the author's response, some of the reviewers' questions and suggestions remain. In particular, the reviewers note that the manuscript had not been updated and that not all of their questions had been answered. (The former isn't required, but can be helpful.) The reviewers were unanimous that this paper was not yet ready to be published at TMLR.

While each reviewer made specific remarks in their recommendations, my sense was the reviewers would have appreciated a more thorough discussion of the results in the paper (even at the cost of increasing the length of the manuscript). Below I provide summaries and some of the salient comments (in quotes) made after the authors' response. Note that I did not consider other comments that discussed the expected impact of the work as these are not part of TMLR's criteria.

- Given LEGO, other types of neural architectures might be able to learn different "shortcuts." Providing additional results either with different architectures or on different tasks would better support the conclusions. Providing results using additional transformer architectures (i.e., in addition to encoder-based ones) would better support the current claim that the conclusions apply to the broad class of Transformers.

- Regarding Figure 4. In the figure "BERT and ALBERT figures are both almost linear, with ALBERT being slightly more linear. This doesn’t necessarily mean that ALBERT is better at reasoning. It is still not clear if the task is solved with only for loop or if having an almost 1-1 relationship between layers and variables mean that a model is implementing a for loop."

- Section 5 could be better integrated with the rest of the paper. In particular, the evaluation protocol in Section 5 is different than with the previous experiments.


I would happily consider an updated resubmission of this work.

**Audience:**

This paper introduces a new reasoning task aimed at transformers. It is of clear interest to the TMLR audience.

**Claims And Evidence:**

The reviewers find that some of the claims might either need rephrasing or require better support. In particular, the authors make claims about the behavior of Transformers from the proposed tasks, and the reviewers find that meaningful empirical comparisons are missing. I have detailed this in the Comments Section below.

**Resubmission Of Major Revision:**

The authors may consider submitting a major revision at a later time.